# Does Chinese-style margin trading promote the high-quality development of listed companies?

**Jingyao Tang** [1]*, **Yu Wu**[2], **Yunqi Ye**[1]

**1** Institute of Chinese Financial Studies, Southwestern University of Finance and Economics, Chengdu, China, **2** Bank of Chongqing, Chongqing, China

\* 284812781@qq.com

## Abstract

Using the margin trading reform of China's stock market as a quasinatural experiment, this paper explores whether margin trading promotes the high-quality development of listed companies. The results show that total factor productivity (TFP) significantly decreases after listed companies are included in the underlying stocks of margin trading. In addition, the negative impacts are stronger for listed companies with higher financial leverage, lower cash asset holdings, lower shareholdings of financial institution investors, and less attention from securities analysts. Further research shows that the negative impacts of margin trading on TFP are closely related to the deterioration of the information environment and the tightening of financing constraints. When listed companies are included in the underlying stocks of margin trading, they use a lower proportion of their net profit for internal financing (and a higher proportion of their net profit for cash dividends) and significantly reduce external equity financing. The results of this study show that the margin trading reform in China's stock market may inhibit the high-quality development of listed companies to a certain extent.

**Data Availability Statement:** All relevant data are within the manuscript and its Supporting information files.

**Funding:** The author(s) received no specific funding for this work.

## Introduction

At present, China's economic goal is shifting from the goal of high-speed growth to that of high-quality development. With the diminishing marginal output of labor and capital inputs, the future development of China's economy will have a greater dependence on the production efficiency and development quality of the real sector. Enterprise innovation, industrial upgrading and the long-term improvement of the production efficiency of the real sector are inseparable from the support of the capital market [1]. As an emerging market, China's stock market construction is still not perfect, and it is significantly different from developed markets in terms of system design, investor structure, policy measures and legal environment. Therefore, only continuous reform according to the actual situation can effectively improve the ability of China's stock market to serve the high-quality development of the real sector. In this context, exploring the best ways to promote the high-quality development of the real sector (i.e., the

**Competing interests:** The authors have declared that no competing interests exist.

total factor productivity (TFP) of real firms) through stock market reform is of great significance for transforming China's economic growth model and stimulating its long-term development momentum.

As an important reform of China's stock market, margin trading was implemented on March 31, 2010, and it gradually expanded the scope of underlying stocks. This means that China's stock market now has formal and legal leverage trading and short-selling channels. In recent years, financial market factors that affect corporate TFP have gradually received more attention [2–4]. However, there are few studies that directly examine how margin trading reform in China's stock market affects the TFP of listed companies. Traditional theory holds that short-selling restrictions cause stock prices to reflect investors' optimism in the short term and lead to mispricing [5]. Therefore, margin trading should remove the short-selling restrictions and improve the pricing efficiency of the stock market. Some studies also posit that short sellers have obvious information advantages, and margin trading promotes short-selling activities, which reduces asymmetric information and stock price volatility and improves stock liquidity and pricing efficiency [6–9]. However, an increasing number of studies show that margin trading in China's stock market increases speculative activities, investor sentiment and the optimistic bias of analysts, which decrease the pricing efficiency of the stock market [10–12]. These contrasting empirical results make it difficult to clarify whether margin trading reform can promote the high-quality development of listed companies. This is also why we focus on the impact of Chinese-style margin trading on the TFP of listed companies.

Through the stock market information environment and corporate financing behavior, we establish a theoretical link between margin trading reform and the TFP of listed companies. On the one hand, according to asset pricing theory, the information asymmetry problem significantly increases the cost of equity financing and reduces the market value and external equity financing motivation of firms [13–16]. On the other hand, according to the dividend signal model, information asymmetry forces companies to increase cash dividends to release positive signals to investors and enhance market value [17–19]. Principal-agent theory also shows that when information is asymmetric, shareholders are motivated to reduce retained earnings as internal financing to reduce the free cash flow at the disposal of managers, thus reducing principal-agent costs and encroaching on the interests of creditors [20, 21]. Therefore, because Chinese-style margin trading essentially encourages speculation and combats short-selling arbitrage [10–12], the information environment of the stock market deteriorates, information asymmetry intensifies, and internal and external financing of listed companies are suppressed. This means that companies may be forced to reduce innovation activities and reduce market share due to a tightening of financing constraints [22, 23]. As a result, the TFP of listed companies will be negatively impacted.

From the perspective of the stock market information environment and firm financing constraints, this paper examines how margin trading reform in China's stock market affects the TFP of listed companies. We take the gradual expansion of the margin trading scope as a quasinatural experiment and use the time-varying propensity score matching difference in difference (PSM-DID) method to study how firms' TFP changes after including the scope of margin trading. The results show that the implementation of margin trading reform exerts significantly negative impacts on the TFP of those firms within the scope of margin trading. The negative impacts of margin trading reform are stronger in companies with higher financial leverage, lower cash asset holdings, lower shareholdings of financial institution investors, and less attention from securities analysts. Further research shows that these negative effects are closely related to the deterioration of the stock market information environment and the tightening of corporate financing constraints. On the one hand, the deterioration of the

information environment causes companies to tend to use higher portions of net profits to pay cash dividends rather than using internal financing. On the other hand, the deterioration of the information environment also increases the cost of equity financing for listed companies, thereby reducing the scale of external equity financing. Our results show how margin trading affects the financing decisions of listed companies by changing the stock market information environment, and ultimately transmitting it to TFP.

The marginal contributions made by this paper are as follows: (I) The extant literature places less of a focus on whether margin trading can promote the high-quality development and TFP of listed companies. By examining the effect of China's stock market margin trading reform on the TFP of listed companies, this paper further enriches the research on margin trading and provides a new perspective from which to evaluate China's stock market margin trading reform policy. (II) The existing literature pays more attention to the impact of margin trading on the quality of the stock market, but few studies assess how margin trading affects the business decisions and business performance of listed companies. From the perspective of the stock market information environment and corporate financing constraints, this paper links margin trading reform with the high-quality development of the real sector and thus provides an important reference for the best means of promoting the TFP growth and high-quality development of listed companies by deepening China's capital market reform. (III) Margin trading can theoretically break market short-selling restrictions, thereby easing information asymmetry and improving pricing efficiency. However, the results of this paper show that Chinese-style margin trading does not play a role in breaking short-selling restrictions but rather worsens the information environment. Therefore, the margin trading reform of China's stock market does not promote the high-quality development of listed companies but rather inhibits the growth of TFP to a certain extent. The conclusion of this paper has theoretical and practical significance for the further improvement of the margin trading system in China's stock market.

The rest of this paper is arranged as follows: The second section presents the literature review and research hypothesis; the third section details the empirical research design; the fourth section contains the empirical results and analysis; the fifth section presents the extension research; and the sixth section concludes.

## Literature review and research hypothesis

The literature on developed stock markets generally claims that short selling is conducive to promoting price discovery, improving stock liquidity and reducing earnings management, thereby improving stock market pricing efficiency and increasing firm-level market value [24–30]. However, recent studies on China's stock market show that margin trading leads to investors' overoptimism, exacerbates analysts' optimism bias, increases insider trading behavior, and exacerbates stock price volatility and stock price crash risk, which reduces stock pricing efficiency [10–12]. This may be related to the following factors. First, the asymmetric scale of long-buying transactions and short-selling transactions in China's stock market and speculative behaviors are amplified by margin trading. This causes margin trading to not only fail to promote short-selling arbitrage but also drives the excessive optimism of investors and increases noise trading [31]. The ratio of short-selling to total trading volume in China's stock market is approximately 0.43%, which is much lower than the 12.9% level of the New York Stock Exchange [25]. Second, the selection criteria for the underlying stocks of margin trading ensure that firms have better information disclosure quality and business performance in margin trading scope, which makes it difficult for the short-selling mechanism to play a role [31]. Third, the institutional construction of China's stock market is not yet optimal, its stock price

manipulation problems are more serious, and they are driven by controlling shareholders and a weaker protection of investor rights [23].

When margin trading causes the stock market information environment to deteriorate and stock pricing efficiency to decline, those firms included in the margin trading scope might face increased asymmetric information and tightened financing constraints as a result. On the one hand, margin trading may lead companies to increase their cash dividends and reduce their level of earnings retention, which implies a reduction in internal financing. According to principal-agent theory, an increase in the level of information asymmetry intensifies principal-agent conflicts, and shareholders become more likely to increase their cash dividends and reduce their level of earnings retention. This can increase the occupation of creditor funding and reduce agency costs by weakening creditors' claims and reducing managers' disposable free cash flow [20, 21]. At the same time, according to the dividend signal model, a company's increase in cash dividends under asymmetric information conditions can transmit a positive signal to the market regarding the company's expected cash flow, thereby enhancing both the company's market value and shareholders' wealth [17–19]. On the other hand, margin trading may lead companies to reduce or refuse external equity financing. When managers or their potential counterparts have more information advantages, stock investors reduce their participation in trading or demand higher expected returns to compensate for the risk of information asymmetry in trading. This, in turn, increases the cost of equity financing [13–16]. Meanwhile, under the condition of information asymmetry, companies issuing new shares for financing may release adverse signals to the market, indicating that stock prices are overvalued, so they tend to reduce or refuse to issue new shares for external financing [22, 23].

The tightening of financing constraints inevitably exerts a negative impact on the company's TFP. First, under the condition of information asymmetry, the tightening of financing constraints may lead companies to abandon good investment opportunities, especially those companies with less cash flow and less debt issuance potential, as well as those conducting large upfront capital investment projects. This makes the company's actual investment level lower than expected under full information conditions with no financing constraints [19]. Second, high equity financing costs and insufficient external equity financing also inhibit a company's innovative R&D activities. Because innovative R&D activities have the characteristics of large initial capital investment, slow formation of productivity and high uncertainty of future cash flow, when the financial impact caused by the tightening of corporate financing constraints exceeds the buffer capacity of its working capital, the company is forced to reduce its level of innovation R&D investment, thus inhibiting the increase in TFP [1]. Third, financing constraints may also cause companies to delay or abandon the upgrading of existing production equipment. Finally, companies that lack sufficient equity capital have difficulty gaining scale economy and specialization advantages quickly, so it is a challenge to gain an advantage in long-term market competition. In summary, this paper develops the following hypothesis:

**Hypothesis: Chinese-style margin trading negatively impacts the TFP of listed companies by increasing information asymmetry and financing constraints**.

## Empirical research design

### Model setup

The margin trading reform in China's stock market is an exogenous policy shock. Moreover, it gradually expands the scope of margin trading underlyings (Table 1 reports the previous

**Table 1. Expansions of underlying stocks of margin trading in China's A-share market.**

| Time | Expansion batch | Number of new underlying stocks | Number of underlying stocks after expansion |
|---|---|---|---|
| March 2010 | | | 90 |
| October 2011 | 1 | 195 | 285 |
| January 2013 | 2 | 215 | 500 |
| September 2013 | 3 | 200 | 700 |
| September 2014 | 4 | 200 | 900 |
| December 2016 | 5 | 50 | 950 |
| August 2019 | 6 | 650 | 1600 |

expansion of margin trading underlyings in the A-share market), which well meets the conditions for using a time-varying difference-in-difference (DID) model [32]. Therefore, this paper constructs the following time-varying DID panel data model to examine the effects of margin trading on the TFP of listed companies:

$$TFP_{i,t} = \alpha + \beta Margin_{i,t} + \sum_{k=1}^{K} \gamma_k Control_{k,i,t} + u_i + v_t + \varepsilon_{i,t} \qquad (1)$$

where the dependent variable $TFP_{i,t}$ is the total factor productivity of listed company $i$ in year $t$; the independent variable $Margin_{i,t}$ is a dummy variable for DID analysis; the value of company $i$ when included in the underlying stock margin trading from year $t + 1$ is 1, and it is 0 otherwise; $Control_{k,i,t}$ indicate control variables, and $K$ is the number of control variables; $u_i$ and $v_t$ are dummy variables for firm and year fixed effects, respectively; $\varepsilon_{i,t}$ is the residual error; $\alpha$ is a constant term; and $\beta$ and $\gamma_k$ are regression coefficients. We focus on the regression coefficient $\beta$, which represents the difference in TFP between the experimental group ($Margin_{i,t} = 1$) and the control group ($Margin_{i,t} = 0$). In accordance with the hypothesis of this paper, we expect that the estimated value of the regression coefficient $\beta$ in Model (1) should be significantly negative, i.e., that margin trading essentially reduces the TFP level of the underlying listed companies. In other words, the net effect of margin trading reform on the TFP of listed companies is significantly negative.

The model has the following advantages. First, the large panel data sample contains both cross-sectional dimension and time series dimension information, which is conducive to improving the accuracy of model estimation. Second, the DID model helps in controlling potential endogeneity problems, and thus can better identify the net effect following the implementation of the policy. Third, the inclusion of company and year fixed effects in the model is conducive to alleviating potential missing variables.

In addition, three aspects need to be considered. First, to alleviate the potential sample selection bias between the experimental group and the control group and improve their sample randomness, this paper also conducts propensity score matching (PSM) on the sample data before DID estimation in Model (1).

Second, to further ensure the reliability of the empirical results, this paper uses three different methods to estimate the TFP level of companies. The first is based on the OP method [33], which is a semiparametric two-step estimation method that uses investment as a proxy variable for company productivity. The second is based on the LP method [34] and uses intermediate products as a semiparametric estimation method for the proxy variable indicating company productivity, which can overcome the problem caused by the OP method not being able to be

**Table 2. Variable description.**

| Variables | Description |
|-----------|-------------|
| OP | Firm TFP estimated by the OP method |
| LP | Firm TFP estimated by the LP method |
| WRDG | Firm TFP estimated by the GMM method |
| Margin | Dummy variable noted as 1 when the company is included in the underlying stock of margin trading and 0 otherwise |
| Size | Natural logarithm of total assets |
| Leverage | Ratio of total liabilities to total assets |
| NCA | Ratio of noncurrent assets to total assets |
| CapInt | Ratio of fixed assets to employee salary |
| Growth | Operating income year-on-year growth rate |
| ROAVOL | Standard deviation of return on total assets in recent 3 years |
| OPAQ | Manipulable accrued surplus |

estimated when the investment amount is 0. The third is based on the GMM method [35] and is a one-step estimation method that can also overcome the potential identification problem in the semiparametric estimation method.

Third, this paper also selects the following indicators as covariates or control variables of the model: (I) the natural logarithm of total assets, *Size*, is used to control the size of the company; (II) the ratio of total liabilities to total assets, *Leverage*, is used to control the company's capital structure; (III) the ratio of noncurrent assets to total assets, *NCA*, is used to control the company's asset structure; (IV) the ratio of net fixed assets to employee compensation, *CapInt*, is used to control the company's capital density or factor input portfolio; (V) operating income year-on-year growth rate, *Growth*, is used to control the company's growth cycle or future investment opportunities; (VI) the standard deviation of the return on total assets over the past three years, *ROAVOL*, is used to control the company's operating risks; and (VII) operable accruals, *OPAQ* [36], is used to control the quality of corporate information disclosure. Table 2 summarizes and explains the main variables in the empirical research.

## Data

Chinese A-share listed companies over the period of 2007 to 2021 are used as a research sample in this study. Referring to the literature, the sample data are preprocessed as follows: (I) all ST companies are eliminated; (II) all financial industry companies (according to the 2012 edition of the China Securities Regulatory Commission industry classification criteria) are excluded; (III) samples with missing values in the main transaction information or financial information are eliminated; (IV) samples with less than 2 years of listing are excluded; (V) samples with asset-liability ratio greater than 100% are eliminated; (VI) companies that have never been included in the underlying stocks during the sample period are excluded; and (VII) to mitigate the impact of extreme data, all continuous variables are winsorized at the 1% and 99% levels. After processing, a total of 8629 sets of firm-year observation samples were obtained. It is worth noting that due to the reform of China's accounting standards for firms in 2006, the sample period begins in 2007 to exclude the impact of these changes. Table 3 reports the descriptive statistical results of the main variables. The data source is the CSMAR database.

**Table 3. Descriptive statistics.**

| Variables | N | Mean | S.D. | Min | Median | Max |
|---|---|---|---|---|---|---|
| OP | 8629 | 10.421 | 1.002 | 5.089 | 10.446 | 15.750 |
| LP | 8629 | 10.090 | 0.998 | 4.610 | 10.140 | 14.671 |
| WRDG | 8629 | 9.697 | 0.990 | 4.194 | 9.748 | 14.328 |
| Margin | 8629 | 0.428 | 0.495 | 0.000 | 0.000 | 1.000 |
| Size | 8629 | 22.817 | 1.277 | 19.117 | 22.684 | 25.928 |
| Lev | 8629 | 0.460 | 0.192 | 0.053 | 0.465 | 0.987 |
| NCA | 8629 | 0.450 | 0.205 | 0.038 | 0.438 | 0.907 |
| CapInt | 8629 | 3.203 | 1.366 | 0.258 | 3.102 | 7.685 |
| Growth | 8629 | 0.247 | 0.455 | -0.657 | 0.157 | 3.261 |
| ROAVOL | 8629 | 0.020 | 0.029 | 0.001 | 0.013 | 0.397 |
| OPAQ | 8629 | 0.054 | 0.056 | 0.001 | 0.038 | 0.353 |

## Basic empirical facts

Table 4 reports the proportion of buy-to-let and short-sell transactions in the underlying stocks of A-share margin trading during 2010–2021. The proportion of long-buying refers to the ratio of the year of the underlying company's long-buying amount and the total transaction amount, and short-selling transactions refers to the ratio of the year of the underlying company's short-selling shares and the total number of shares traded. The last column of the table shows the ratio of the proportion of long-buying to the proportion of short-selling transactions in the year. It is not difficult to see that thus far, the margin trading in the A-share market is highly concentrated in the buy-to-let category, and the proportion of buy-to-let transactions is much higher than the proportion of short-selling transactions—the highest ratio of the two reached 282.31 times (in 2016), the lowest reached as high as 5.49 times (in 2012), and the average was 31.88 times. Although the ratio showed a downward trend after 2016, it could not change the speculative frenzy of the A-share market that followed the margin trading reform. This empirical fact means that short-selling arbitrage activities in the A-share market could face high noise trading risks and in fact serve to increase arbitrage restrictions rather than to break short-selling restrictions.

**Table 4. Proportion of margin trading.**

| Year | Proportion of long-buying | Proportion of short-selling | Long-buying/short-selling |
|---|---|---|---|
| 2010 | 0.67% | 0.01% | 64.47 |
| 2011 | 1.39% | 0.18% | 7.90 |
| 2012 | 6.63% | 1.21% | 5.49 |
| 2013 | 9.60% | 1.01% | 9.49 |
| 2014 | 17.35% | 0.98% | 17.71 |
| 2015 | 20.10% | 0.79% | 25.46 |
| 2016 | 17.86% | 0.06% | 282.31 |
| 2017 | 18.05% | 0.17% | 104.65 |
| 2018 | 15.97% | 0.27% | 59.92 |
| 2019 | 11.28% | 0.14% | 78.46 |
| 2020 | 12.33% | 0.27% | 44.92 |
| 2021 | 10.80% | 0.38% | 28.16 |
| Total | 13.58% | 0.43% | 31.88 |

## Empirical results and analysis

### PSM results

As mentioned above, to ensure that the experimental group and the control group have sample randomness in the time-varying DID model and to avoid sample selection errors, this paper uses a caliper matching method with a radius of 0.05 to perform PSM processing on the sample data in advance. Table 5 reports the results of the PSM processing. After PSM processing, the standard percentage bias of each covariate is significantly reduced, and this reduction is all within 10% (mostly within 5%). This shows that PSM processing effectively alleviates any potential sample selection bias and gives the experimental group and the control group good comparability. In addition, after matching, the number of samples in the experimental group and the control group is 3682 and 4889, respectively, for a total of 8571 sample observations, which also better meets the conditions for using large samples.

### DID results

Table 6 reports the PSM-DID estimation results of Model (1). The results show that in all regressions, the estimated regression coefficient of the explanatory variable *Margin* is negative at the 1% significance level, even after adding control variables. This shows that margin trading reform has a statistically significant negative impact on the TFP of listed companies. In addition, according to the size of the estimated value of the regression coefficient, when a company is included in the underlying margin trading, its TFP level decreases by approximately 0.22. According to descriptive statistics, the average TFP in the sample is approximately 10. Therefore, the margin trading reform leads to a decrease in the TFP of the underlying company by approximately 2% compared with that in the prereform period. From an economic perspective, this negative impact cannot be ignored. The above empirical results are completely consistent with the expectations of this study, which verifies the hypothesis claiming that the

**Table 5. PSM results.**

| Covariates | Before matching: U | Sample mean | | Standard percentage bias |
|---|---|---|---|---|
| | After matching: M | Experimental group | Control group | |
| *Size* | U | 23.469 | 22.330 | 98.7 |
| | M | 23.460 | 23.467 | -0.6 |
| *Lev* | U | 0.471 | 0.453 | 9.5 |
| | M | 0.471 | 0.471 | 0.2 |
| *NCA* | U | 0.451 | 0.449 | 0.9 |
| | M | 0.451 | 0.457 | -2.9 |
| *CapInt* | U | 3.050 | 3.318 | -19.7 |
| | M | 3.051 | 3.057 | -0.5 |
| *Growth* | U | 0.167 | 0.306 | -31.6 |
| | M | 0.168 | 0.194 | -5.9 |
| *ROAVOL* | U | 0.018 | 0.022 | -14.1 |
| | M | 0.018 | 0.020 | -6.2 |
| *OPAQ* | U | 0.047 | 0.060 | -25 |
| | M | 0.047 | 0.048 | -3.2 |
| | Unmatched number | Matched number | Total | |
| Control group | 44 | 4,889 | 4,933 | |
| Experimental group | 14 | 3,682 | 3,696 | |
| Total | 58 | 8,571 | 8,629 | |

**Table 6. The impact of margin trading on TFP.**

| | (1) | (2) | (3) | (4) | (5) | (6) |
|---|---|---|---|---|---|---|
| | *OP* | *OP* | *LP* | *LP* | *WRDG* | *WRDG* |
| *Margin* | -0.219*** | -0.224*** | -0.220*** | -0.226*** | -0.222*** | -0.227*** |
| | (-6.59) | (-7.25) | (-6.79) | (-7.43) | (-6.85) | (-7.43) |
| *Size* | | 0.304*** | | 0.305*** | | 0.291*** |
| | | (8.99) | | (9.06) | | (8.64) |
| *Leverage* | | -0.434*** | | -0.465*** | | -0.464*** |
| | | (-3.34) | | (-3.59) | | (-3.58) |
| *NCA* | | -0.882*** | | -0.939*** | | -0.959*** |
| | | (-6.32) | | (-6.75) | | (-6.87) |
| *CapInt* | | 0.130*** | | 0.051** | | 0.054*** |
| | | (6.10) | | (2.46) | | (2.59) |
| *Growth* | | 0.193*** | | 0.197*** | | 0.196*** |
| | | (10.06) | | (10.22) | | (10.14) |
| *ROAVOL* | | -0.139 | | -0.068 | | -0.072 |
| | | (-0.31) | | (-0.15) | | (-0.16) |
| *OPAQ* | | 0.577*** | | 0.580*** | | 0.584*** |
| | | (3.55) | | (3.57) | | (3.58) |
| Constant | 10.041*** | 3.596*** | 9.698*** | 3.532*** | 9.326*** | 3.452*** |
| | (184.22) | (4.87) | (180.50) | (4.83) | (173.92) | (4.71) |
| Firm FE | Yes | Yes | Yes | Yes | Yes | Yes |
| Year FE | Yes | Yes | Yes | Yes | Yes | Yes |
| Robust | Yes | Yes | Yes | Yes | Yes | Yes |
| N | 8571 | 8571 | 8571 | 8571 | 8571 | 8571 |
| R2 | 0.156 | 0.225 | 0.174 | 0.239 | 0.165 | 0.228 |

Note: ***, **, and * indicate significance at the 1%, 5% and 10% levels, respectively. The values in parentheses represent t values based on robust standard errors.

reform of margin trading in China's A-share market exerts a significant negative impact on the TFP of listed companies. This means that the reform of margin trading in China's stock market does not promote the high-quality development of listed companies but rather restricts their TFP growth.

## Dynamic analysis and parallel trend test

One of the important prerequisites of the DID model is that the experimental group and the control group samples must meet the parallel trend hypothesis. Therefore, this article also conducted a parallel trend test based on the event study method on the experimental group and the control group samples. At the same time, the model can also dynamically analyze the time series characteristics of the effect of margin trading on TFP. The measurement model is as follows:

$$TFP_{i,t} = \alpha + \sum_{p=-5}^{5} \beta_p Margin_{i,t} \times Time_{i,p} + \sum_{k=1}^{K} \gamma_k Control_{k,i,t} + u_i + v_t + \varepsilon_{i,t} \qquad (2)$$

where $p = t - l, p \in [-5,5]$, and $l$ represents the year that the company was included in the underlying stock of margin trading. Therefore, $p$ indicates that $t$ is either the year before or after the shock. $Time_{i,p}$ is the dummy variable corresponding to $p$. For example, for company $i$,

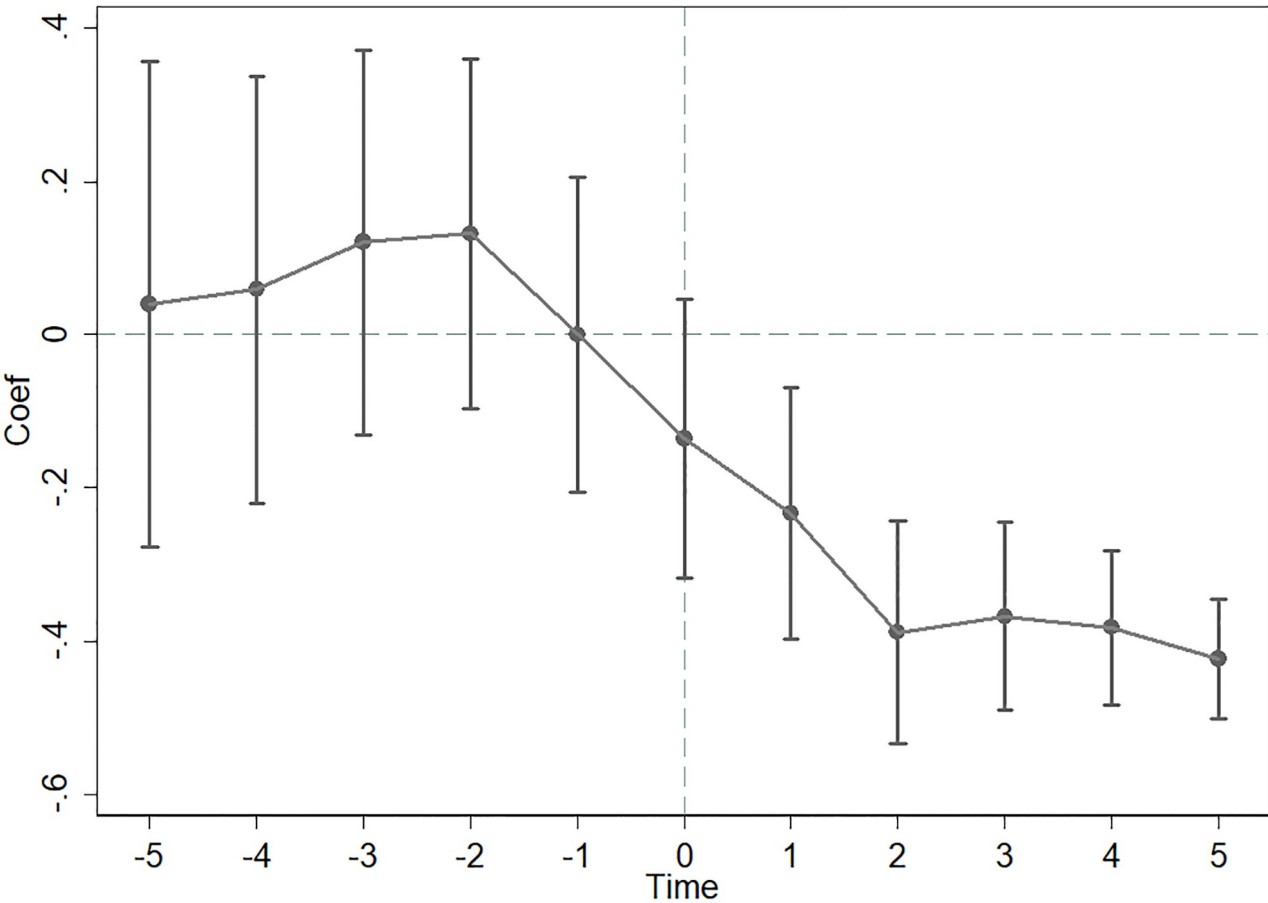

**Fig 1. Dynamic analysis and parallel trend test.**

when $p = -5$ (5 years before the shock to the company), $Time_{i,-5} = 1$; otherwise, $Time_{i,-5} = 0$. The regression coefficient $\beta_p$ reflects the difference in TFP between the treatment and control groups over time. According to the parallel trend assumption, $\beta_p$ should not significantly differ from 0 before the shock but should be significantly negative after the shock.

Fig 1 shows the estimation results of model (2). The vertical axis represents the estimated value of coefficient $\beta_p$ and its confidence interval at the 95% level, and the horizontal axis represents the time variable $Time_{i,p}$. The results show that before the shock, the difference in TFP between the experimental group and the control group is approximately zero, while after the shock, the TFP difference between the experimental group and the control group is significantly negative. This shows that the experimental group and the control group samples meet the parallel trend hypothesis of the DID method. In addition, the results of Fig 1 also reflect the dynamic impact of margin trading on the underlying company throughout a time series. It is not difficult to see that the impact of margin trading on TFP has strong continuity throughout time, rather than simply exerting a temporary impact.

## Cross-section analysis

According to the perspective taken in this paper, margin trading reduces the company's TFP by deteriorating the information environment and tightening equity financing constraints.

Table 7. Cross-sectional analysis by financial leverage.

| | Low Financial Leverage | | | High Financial Leverage | | |
|---|---|---|---|---|---|---|
| | (1) | (2) | (3) | (4) | (5) | (6) |
| | *OP* | *LP* | *WRDG* | *OP* | *LP* | *WRDG* |
| *Margin* | -0.148*** | -0.156*** | -0.156*** | -0.281*** | -0.278*** | -0.279*** |
| | (-4.02) | (-4.28) | (-4.27) | (-5.75) | (-5.81) | (-5.80) |
| Control Variables | Yes | Yes | Yes | Yes | Yes | Yes |
| Firm FE | Yes | Yes | Yes | Yes | Yes | Yes |
| Year FE | Yes | Yes | Yes | Yes | Yes | Yes |
| Robust | Yes | Yes | Yes | Yes | Yes | Yes |
| N | 4291 | 4291 | 4291 | 4280 | 4280 | 4280 |
| R2 | 0.223 | 0.246 | 0.235 | 0.241 | 0.248 | 0.238 |

Note: ***, **, and * indicate significance at the 1%, 5% and 10% levels, respectively. The values in parentheses represent t values based on robust standard errors.

Therefore, in listed companies with a higher risk of financial distress and more serious information asymmetry, the negative impact of margin trading on TFP should be stronger. To verify the above inference, we divide the sample into two subsamples, namely, listed companies with a higher risk of financial distress or more serious information asymmetry and listed companies with a lower risk of financial distress or less information asymmetry.

**Different levels of financial leverage.** First, we assess whether there is cross-sectional heterogeneity among listed companies with different levels of financial leverage. Although financial leverage may provide the benefits of tax shielding, it also increases a company's equity risk level and financial distress cost [37]. More generally, the higher the financial leverage ratio is, the more principal and interest a company needs to repay, the higher the cost of financing through the discovery of new shares, and the smaller the potential for financing through the issuance of new bonds. Therefore, with an increase in financial leverage, companies usually face tighter financing constraints. The negative impact of margin trading on TFP should be stronger in companies with higher financial leverage. We sort the sample companies according to their average asset-liability ratio, equally divide them into two subsamples, and then re-estimate Model (1) on the basis of these two subsamples.

Table 7 reports the estimated results. The results show that, on the one hand, the estimated regression coefficient of the explanatory variable *Margin* is negative at the 1% significance level in all regressions, indicating that the main empirical results of this paper are still valid across companies with different asset-liability ratios. On the other hand, in the subsample with the higher asset-liability ratio, the absolute value of the regression coefficient of the explanatory variable *Margin* is significantly greater than that of the subsample with the lower asset-liability ratio. This shows that the negative impact of margin trading on TFP is indeed stronger in companies with higher financial leverage.

**Different levels of cash holding ratio.** Second, we examine whether there is cross-sectional heterogeneity among companies with different cash holding ratios. Since a higher proportion of cash holdings implies lower external financing needs and a lower risk of financial distress [38], the negative impact of margin trading on TFP should be stronger in companies with low cash holdings. We sort the sample companies according to their average cash holding ratio, equally divide them into two subsamples, and then re-estimate Model (1) on the basis of these two subsamples.

Table 8 reports the estimated results. The results show that in the subsample with lower cash holdings, the absolute value of the regression coefficient of the explanatory variable

**Table 8. Cross-sectional analysis by cash holding ratio.**

| | Low Cash Holding | | | High Cash Holding | | |
|---|---|---|---|---|---|---|
| | (1) | (2) | (3) | (4) | (5) | (6) |
| | *OP* | *LP* | *WRDG* | *OP* | *LP* | *WRDG* |
| *Margin* | -0.247*** | -0.246*** | -0.247*** | -0.192*** | -0.196*** | -0.196*** |
| | (-4.97) | (-5.07) | (-5.07) | (-5.21) | (-5.33) | (-5.33) |
| Control Variables | Yes | Yes | Yes | Yes | Yes | Yes |
| Firm FE | Yes | Yes | Yes | Yes | Yes | Yes |
| Year FE | Yes | Yes | Yes | Yes | Yes | Yes |
| Robust | Yes | Yes | Yes | Yes | Yes | Yes |
| N | 4297 | 4297 | 4297 | 4274 | 4274 | 4274 |
| R2 | 0.192 | 0.206 | 0.196 | 0.283 | 0.298 | 0.287 |

Note: ***, **, and * indicate significance at the 1%, 5% and 10% levels, respectively. The values in parentheses represent t values based on robust standard errors

*Margin* is significantly greater than that of the subsample with higher cash asset holdings, indicating that the negative impact of margin trading on TFP is indeed stronger in companies with lower cash holdings. This further confirms the previous inference.

**Different levels of financial institution shareholding ratio.** Next, we examine whether there is cross-sectional heterogeneity among firms with different proportions of shares held by financial institution investors. On the one hand, a greater number of financial institution investors can improve the information asymmetry between internal and external investors. On the other hand, a company increases its external financing potential by establishing equity linkages with more financial institution investors. Therefore, the negative impact of margin trading on TFP should be stronger in companies with lower shareholdings of financial institution investors. We rank the firms in the sample according to their average shareholding ratio of financial institution investors, equally divide them into two subsamples, and then re-estimate Model (1) on the basis of these the two subsamples.

Table 9 reports the estimated results. The results show that the absolute value of the regression coefficient of the explanatory variable *Margin* is significantly greater than that of the subsample with a higher shareholding ratio of financial institution investors. This shows that the negative impact of margin trading on TFP is indeed stronger in underlying companies with lower shareholding ratios of financial institution investors.

**Table 9. Cross-sectional analysis of different financial institution shareholding ratios.**

| | Low Financial Institution Shareholding Ratio | | | High Financial Institution Shareholding Ratio | | |
|---|---|---|---|---|---|---|
| | (1) | (2) | (3) | (4) | (5) | (6) |
| | *OP* | *LP* | *WRDG* | *OP* | *LP* | *WRDG* |
| *Margin* | -0.310*** | -0.308*** | -0.308*** | -0.111*** | -0.118*** | -0.118*** |
| | (-6.98) | (-7.03) | (-7.03) | (-2.80) | (-2.99) | (-2.99) |
| Firm FE | Yes | Yes | Yes | Yes | Yes | Yes |
| Year FE | Yes | Yes | Yes | Yes | Yes | Yes |
| Robust | Yes | Yes | Yes | Yes | Yes | Yes |
| N | 4287 | 4287 | 4287 | 4284 | 4284 | 4284 |
| R2 | 0.234 | 0.247 | 0.238 | 0.227 | 0.242 | 0.230 |

Note: ***, **, and * indicate significance at the 1%, 5% and 10% levels, respectively. The values in parentheses represent t values based on robust standard errors.

**Table 10. Cross-sectional analysis by different levels of securities analyst attention.**

| | Low Analyst Attention | | | High Analyst Attention | | |
|---|---|---|---|---|---|---|
| | (1) | (2) | (3) | (4) | (5) | (6) |
| | *OP* | *LP* | *WRDG* | *OP* | *LP* | *WRDG* |
| *Margin* | -0.271*** | -0.267*** | -0.268*** | -0.151*** | -0.158*** | -0.158*** |
| | (-5.74) | (-5.75) | (-5.75) | (-4.21) | (-4.43) | (-4.42) |
| Control Variables | Yes | Yes | Yes | Yes | Yes | Yes |
| Firm FE | Yes | Yes | Yes | Yes | Yes | Yes |
| Year FE | Yes | Yes | Yes | Yes | Yes | Yes |
| Robust | Yes | Yes | Yes | Yes | Yes | Yes |
| N | 4296 | 4296 | 4296 | 4275 | 4275 | 4275 |
| R2 | 0.212 | 0.225 | 0.215 | 0.254 | 0.268 | 0.256 |

Note: ***, **, and * indicate significance at the 1%, 5% and 10% levels, respectively. The values in parentheses represent t values based on robust standard errors.

**Different levels of securities analyst attention.** Finally, we examine whether there is cross-sectional heterogeneity among companies that attract different levels of attention from securities analysts. When the company has more securities analysts, external investors can obtain more information about the company's development prospects through the research reports provided by these securities analysts, thus alleviating information asymmetry within the company. Therefore, the negative impact of margin trading on TFP should be more intense in companies that attract a lower level of attention from securities analysts. We rank the companies in the sample according to their average securities analyst attention, divide them equally into two subsamples, and then re-estimate model (1) on the basis of these two subsamples.

Table 10 reports the estimated results. The results show that the absolute value of the regression coefficient of the explanatory variable *Margin* is significantly greater than that of the securities analysts in the subsample of companies with lower levels of attention. This shows that the negative impact of margin trading on TFP is indeed stronger in underlying companies that attract lower levels of attention from securities analysts. The above results verify the inference of this paper.

## Robustness test

To ensure that the main empirical results of this paper are robust and reliable, we conduct the following robustness tests.

(I) Placebo test: Considering that the DID estimation results may be affected by unobservable factors, we disrupt the samples of the experimental group and the control group, rerandomize them into an experimental group and a control group and re-estimate Model (1). By repeating the above operation 500 times, we perform a placebo test on the estimated results in Table 5. Table 11 reports the results of the placebo test. The results show that the regression

**Table 11. Placebo test of model (1).**

| | *OP* | *LP* | *WRDG* |
|---|---|---|---|
| *Margin* | -0.224 | -0.226 | -0.227 |
| P Value | 0.000 | 0.000 | 0.000 |

Note: The null hypothesis of the placebo test is that the regression coefficient estimates of Model (1) are not significantly different from the coefficients obtained by repeated random sampling.

**Table 12. Robustness test—replacement of dependent variables.**

|  | (1) | (2) | (3) | (4) |
|---|---|---|---|---|
|  | *OPacf* | *OPacf* | *LPacf* | *LPacf* |
| *Margin* | -0.238*** | -0.227*** | -0.236*** | -0.226*** |
|  | (-6.99) | (-7.06) | (-6.76) | (-6.93) |
| Control Variables | NO | Yes | No | Yes |
| Firm FE | Yes | Yes | Yes | Yes |
| Year FE | Yes | Yes | Yes | Yes |
| Robust | Yes | Yes | Yes | Yes |
| N | 8571 | 8571 | 8571 | 8571 |
| R2 | 0.064 | 0.132 | 0.061 | 0.139 |

Note: ***, **, and * indicate significance at the 1%, 5% and 10% levels, respectively. The values in parentheses represent t values based on robust standard errors.

coefficient estimates of the explanatory variable Margin in Table 5 are small probability events in the regression coefficient distribution of 500 random sampling estimates (P value is 0.0000). This shows that the possibility of DID estimation results being affected by unobservable factors can be excluded.

(II) Replacement of dependent variables: Considering that the DID estimation results may depend on the TFP estimation method, we replace the dependent variables in Model (1) with the *OPacf* and *LPacf* indicators as adjusted by the ACF method [39] and re-estimate them. Table 12 reports the estimated results. The results are very consistent with the main conclusions of this paper, which shows that the main empirical results of this paper are not sensitive to TFP estimation methods.

(III) Grouped regression by ownership nature: Given the significant differences in business objectives and governance structures between China's state-owned and nonstate-owned listed companies, we divide the sample companies into two subsamples according to whether they are state-owned or not and re-estimate Model (1). Table 13 reports the regression results. The results reveal that regardless of whether a company is state-owned, its TFP is significantly negatively impacted by margin trading.

The results of the above robustness tests show that the main empirical results of this paper are robust and reliable.

**Table 13. Robustness test—regression by property rights.**

|  | State-owned Corporation | | | Nonstate-owned Corporation | | |
|---|---|---|---|---|---|---|
|  | (1) | (2) | (3) | (4) | (5) | (6) |
|  | *OP* | *LP* | *WRDG* | *OP* | *LP* | *WRDG* |
| *Margin* | -0.243*** | -0.242*** | -0.242*** | -0.181*** | -0.187*** | -0.188*** |
|  | (-5.05) | (-5.15) | (-5.14) | (-4.42) | (-4.60) | (-4.61) |
| Control Variables | Yes | Yes | Yes | Yes | Yes | Yes |
| Firm FE | Yes | Yes | Yes | Yes | Yes | Yes |
| Year FE | Yes | Yes | Yes | Yes | Yes | Yes |
| Robust | Yes | Yes | Yes | Yes | Yes | Yes |
| N | 4116 | 4116 | 4116 | 4455 | 4455 | 4455 |
| R2 | 0.233 | 0.240 | 0.231 | 0.232 | 0.253 | 0.241 |

Note: ***, **, and * indicate significance at the 1%, 5% and 10% levels, respectively. The values in parentheses represent t values based on robust standard errors.

## Extension research

### Margin trading, information environment and financing constraints

According to the hypothesis of this paper, the pathway by which the A-share market margin trading reform affects a company's TFP is the deterioration of the information environment and the tightening of financing constraints. To verify whether margin trading has led to a deterioration of the information environment and a tightening of the financing constraints of the underlying company, we use the following econometric model for further research:

$$SYN_{i,t} = \alpha + \beta Margin_{i,t} + \sum_{k=1}^{K} \gamma_k Control_{k,i,t} + u_i + v_t + \varepsilon_{i,t} \tag{3}$$

$$ILLIQ_{i,t} = \alpha + \beta Margin_{i,t} + \sum_{k=1}^{K} \gamma_k Control_{k,i,t} + u_i + v_t + \varepsilon_{i,t} \tag{4}$$

$$MTB_{i,t} = \alpha + \beta Margin_{i,t} + \sum_{k=1}^{K} \gamma_k Control_{k,i,t} + u_i + v_t + \varepsilon_{i,t} \tag{5}$$

$$KZ_{i,t} = \alpha + \beta Margin_{i,t} + \sum_{k=1}^{K} \gamma_k Control_{k,i,t} + u_i + v_t + \varepsilon_{i,t} \tag{6}$$

where $SYN_{i,t}$ is the stock price synchronicity index [21, 40]. The greater the value of the index is, the lower the stock price informativeness. $ILLIQ_{i,t}$ is an illiquidity index [41]. The larger the value of this index is, the worse the liquidity of the stock. $MTB_{i,t}$ is the ratio of a company's market value to book value. The greater that the value of this indicator is, the higher the valuation level of the company's stock. $KZ_{i,t}$ is the financing constraint indicator [42]. The greater the value of this indicator, the tighter the financing constraints that a company faces. According to the hypothesis of this paper, we expect that the estimated values of regression coefficient $\beta$ in Model (3), Model (4) and Model (6) are significantly positive, and that the estimated value of regression coefficient $\beta$ in Model (5) is significantly negative. That is, when listed companies are included in the underlying stocks of margin trading, the information environment deteriorates significantly, the market value of the company decreases significantly, and the relevant financing constraints tighten significantly.

Table 14 reports the estimation results of Model (3)-Model (6). The results show that the regression coefficient estimates of the explanatory variable *Margin* in Model (3), Model (4) and Model (6) are positive at the 1% significance level, regardless of whether control variables have been added, while the regression coefficient estimates of the explanatory variable *Margin* in Model (5) are negative at the 1% significance level. The results are completely consistent with expectations, which shows that when listed companies are included in the underlying stocks of margin trading, their information environment deteriorates significantly, their valuation level is significantly reduced, and their financing constraints are significantly tightened.

### Margin trading and internal financing

According to principal-agent theory and dividend signal theory, under the context of information asymmetry, shareholders have a stronger incentive to increase cash dividends and reduce retained earnings. On the one hand, this can increase the level of shareholder encroachment

**Table 14. Impact of margin trading on the information environment and financing constraints.**

| | Model (3) | | Model (4) | | Model (5) | | Model (6) | |
|---|---|---|---|---|---|---|---|---|
| | (1) | (2) | (3) | (4) | (5) | (6) | (7) | (8) |
| | *SYN* | *SYN* | *ILLIQ* | *ILLIQ* | *MTB* | *MTB* | *KZ* | *KZ* |
| *Margin* | 0.296*** | 0.261*** | 0.008*** | 0.009*** | -0.619*** | -0.496*** | 0.313*** | 0.231*** |
| | (8.40) | (7.39) | (6.18) | (7.76) | (-7.33) | (-6.18) | (5.75) | (4.68) |
| Control Variables | No | Yes | No | Yes | No | Yes | No | Yes |
| Firm FE | Yes | Yes | Yes | Yes | Yes | Yes | Yes | Yes |
| Year FE | Yes | Yes | Yes | Yes | Yes | Yes | Yes | Yes |
| Robust | Yes | Yes | Yes | Yes | Yes | Yes | Yes | Yes |
| N | 8483 | 8483 | 8570 | 8570 | 8570 | 8570 | 8084 | 8084 |
| R2 | 0.426 | 0.438 | 0.313 | 0.346 | 0.265 | 0.302 | 0.044 | 0.149 |

Note: ***, **, and * indicate significance at the 1%, 5% and 10% levels, respectively. The values in parentheses represent t values based on robust standard errors.

on creditor wealth and reduce the disposable free cash flow and agency costs for managers [20, 43]. On the other hand, this can also transmit positive signals to the market about the company's expected cash flow [17–19]. In addition, when listed companies are included in margin trading underlying stocks, their stock liquidity deteriorates significantly. This increases the transaction cost of investors selling stocks to obtain spot cash flow, so investors become more inclined to obtain spot cash flow through an increase in cash dividends.

To test whether A-share listed companies have a tendency to increase their cash dividend payments and reduce their retained earnings (thereby reducing their internal financing) after being included in the underlying stocks of margin trading, this paper constructs the following econometric model:

$$Div_{i,t} = \alpha + \beta Margin_{i,t} + \sum_{k=1}^{K} \gamma_k Control_{k,i,t} + u_i + v_t + \varepsilon_{i,t} \tag{7}$$

$$InFin_{i,t} = \alpha + \beta Margin_{i,t} + \sum_{k=1}^{K} \gamma_k Control_{k,i,t} + u_i + v_t + \varepsilon_{i,t} \tag{8}$$

where $Div_{i,t}$ is the cash dividend payment rate, i.e., the ratio of cash dividend to net profit for the year. The larger that this value is, the greater the proportion of the company's net profit that is used to pay cash dividends to shareholders. $InFin_{i,t}$ is the ratio of retained earnings to net profit for the year. The smaller that this value is, the smaller the proportion of the company's net profit used to support internal financing. According to the hypothesis of this paper, we expect that the estimated value of regression coefficient $\beta$ in Model (7) should be significantly positive, and the estimated value of regression coefficient $\beta$ in Model (8) should be significantly negative. That is, margin trading significantly increases the cash dividend payment ratio of an underlying company, significantly reduces its earnings retention ratio, and inhibits the company's internal financing through its own surplus.

Table 15 reports the estimated results of Model (7) and Model (8). The results show that the regression coefficient estimates of the explanatory variable *Margin* in Model (7) are positive at a significance level of 5%, and the regression coefficient estimates of the explanatory variable *Margin* in Model (8) are negative at a significance level of 5%, after control variables have been added. These results are consistent with expectations, indicating that margin trading has

**Table 15. Impact of margin trading on internal financing.**

|  | Model(7) | | Model(8) | |
| --- | --- | --- | --- | --- |
|  | (1) | (2) | (3) | (4) |
|  | *Div* | *Div* | *InFin* | *InFin* |
| *Margin* | 0.026** | 0.024** | -0.026** | -0.024** |
|  | (2.51) | (2.29) | (-2.51) | (-2.29) |
| Control Variables | No | Yes | No | Yes |
| Firm FE | Yes | Yes | Yes | Yes |
| Year FE | Yes | Yes | Yes | Yes |
| Robust | Yes | Yes | Yes | Yes |
| N | 8291 | 8291 | 8291 | 8291 |
| R2 | 0.017 | 0.035 | 0.017 | 0.035 |

Note: ***, **, and * indicate significance at the 1%, 5% and 10% levels, respectively. The values in parentheses represent t values based on robust standard errors.

indeed significantly increased the cash dividend payment ratio of the underlying company while significantly reducing its income retention ratio and inhibiting the company's internal financing through its own earnings. This implies that the company will use a smaller proportion of operating profits for internal financing, thereby expanding its external financing needs and tightening equity financing constraints.

## Margin trading and external equity financing

According to asset pricing theory, when informational asymmetry in the stock market increases, investors demand a higher expected rate of return as compensation for the risk of information asymmetry that accompanies the holding of that company's shares. This obviously increases the cost of equity financing for companies [13–16]. Therefore, when A-share listed companies are included in the underlying stocks of margin trading, their external equity financing scale is also suppressed due to increased costs. To verify this, this paper constructs the following measurement model:

$$Equity_{i,t} = \alpha + \beta Margin_{i,t} + \sum_{k=1}^{K} \gamma_k Control_{k,i,t} + u_i + v_t + \varepsilon_{i,t} \tag{9}$$

$$Debt_{i,t} = \alpha + \beta Margin_{i,t} + \sum_{k=1}^{K} \gamma_k Control_{k,i,t} + u_i + v_t + \varepsilon_{i,t} \tag{10}$$

where $Equity_{i,t}$ is the natural logarithm of the net cash flow of a company's external equity financing. The larger that this value is, the larger the scale of the company's external equity financing. $Debt_{i,t}$ is the natural logarithm of the net cash flow of the company's external debt financing. The larger that this value is, the larger the scale of the company's external debt financing. In addition, to avoid potential endogeneity problems, the control variables in Model (9) and Model (10) do not include a company's asset-liability ratio, or *Leverage*. According to the inference, we expect that the estimated value of the regression coefficient $\beta$ in Model (9) should be significantly negative. That is, we expect that margin trading will have significantly reduced the size of the underlying company's external equity financing. Model (10) is used as a control. If the estimated value of the regression coefficient $\beta$ in Model (10) is significantly regular, it means that when external equity financing is inhibited, external debt

**Table 16. Impact of margin trading on external equity financing.**

| | Model(9) | | Model(10) | |
|---|---|---|---|---|
| | (1) | (2) | (3) | (4) |
| | *Equity* | *Equity* | *Debt* | *Debt* |
| *Margin* | -0.825** | -0.666** | -0.073 | -0.257 |
| | (-2.48) | (-1.99) | (-0.36) | (-1.32) |
| Control Variables | No | Yes | No | Yes |
| Firm FE | Yes | Yes | Yes | Yes |
| Year FE | Yes | Yes | Yes | Yes |
| Robust | Yes | Yes | Yes | Yes |
| N | 8556 | 8556 | 8571 | 8571 |
| R2 | 0.025 | 0.030 | 0.022 | 0.062 |

Note: ***, **, and * indicate significance at the 1%, 5% and 10% levels, respectively. The values in parentheses represent t values based on robust standard errors.

financing is a better alternative. Otherwise, it means that external debt financing cannot make up for the financing gap caused by the reduction in external equity financing, and the company's financing constraints will be tightened with the reduction in external equity financing.

Table 16 reports the estimated results of Model (9) and Model (10). The results show that the regression coefficient estimates of the explanatory variable *Margin* in Model (9) are negative at the 5% significance level, regardless of whether or not control variables have been added. This shows that margin trading indeed significantly reduces the scale of external equity financing of an underlying company. The regression coefficient estimates of the explanatory variable *Margin* in model (10) are approximately zero at the 10% significance level. This shows that external debt financing cannot make up for the financing gap caused by the reduction in external equity financing, and a company's financing constraints tighten with a reduction in external equity financing. The empirical results are consistent with expectations.

## Financing constraints and TFP

Finally, we continue to examine whether a tightening of financing constraints (including the reduction in internal and external equity financing) has a significant negative impact on a company's TFP. The measurement model is as follows:

$$TFP_{i,t} = \alpha + \beta InFin_{i,t} + \sum_{k=1}^{K} \gamma_k Control_{k,i,t} + u_i + v_t + \varepsilon_{i,t} \tag{11}$$

$$TFP_{i,t} = \alpha + \beta Equity_{i,t} + \sum_{k=1}^{K} \gamma_k Control_{k,i,t} + u_i + v_t + \varepsilon_{i,t} \tag{12}$$

$$TFP_{i,t} = \alpha + \beta KZ_{i,t} + \sum_{k=1}^{K} \gamma_k Control_{k,i,t} + u_i + v_t + \varepsilon_{i,t} \tag{13}$$

According to the hypothesis of this paper, we expect that the estimated value of the regression coefficient $\beta$ in Model (11) and Model (12) should be significantly positive. That is, the increase in corporate TFP is significantly dependent on the scale of both internal financing and external equity financing. When the levels of internal financing and external equity

**Table 17. Impact of financing constraints on TFP.**

| | Model (11) | | | | | |
|---|---|---|---|---|---|---|
| | (1) | (2) | (3) | (4) | (5) | (6) |
| | OP | OP | LP | LP | WRDG | WRDG |
| InFin | 0.381*** | 0.407*** | 0.391*** | 0.418*** | 0.389*** | 0.417*** |
| | (8.86) | (10.04) | (9.24) | (10.30) | (9.20) | (10.27) |
| Control Variables | No | Yes | No | Yes | No | Yes |
| Firm FE | Yes | Yes | Yes | Yes | Yes | Yes |
| Year FE | Yes | Yes | Yes | Yes | Yes | Yes |
| Robust | Yes | Yes | Yes | Yes | Yes | Yes |
| N | 8291 | 8291 | 8291 | 8291 | 8291 | 8291 |
| R2 | 0.193 | 0.271 | 0.215 | 0.286 | 0.204 | 0.274 |
| | Model (12) | | | | | |
| | (1) | (2) | (3) | (4) | (5) | (6) |
| | OP | OP | LP | LP | WRDG | WRDG |
| Equity | 0.006*** | 0.007*** | 0.006*** | 0.007*** | 0.006*** | 0.007*** |
| | (6.55) | (7.59) | (6.63) | (7.82) | (6.57) | (7.74) |
| Control Variables | No | Yes | No | Yes | No | Yes |
| Firm FE | Yes | Yes | Yes | Yes | Yes | Yes |
| Year FE | Yes | Yes | Yes | Yes | Yes | Yes |
| Robust | Yes | Yes | Yes | Yes | Yes | Yes |
| N | 8556 | 8556 | 8556 | 8556 | 8556 | 8556 |
| R2 | 0.152 | 0.222 | 0.171 | 0.236 | 0.161 | 0.226 |
| | Model (13) | | | | | |
| | (1) | (2) | (3) | (4) | (5) | (6) |
| | OP | OP | LP | LP | WRDG | WRDG |
| KZ | -0.159*** | -0.181*** | -0.166*** | -0.186*** | -0.166*** | -0.185*** |
| | (-17.72) | (-20.11) | (-18.95) | (-21.15) | (-18.94) | (-20.96) |
| Control Variables | No | Yes | No | Yes | No | Yes |
| Firm FE | Yes | Yes | Yes | Yes | Yes | Yes |
| Year FE | Yes | Yes | Yes | Yes | Yes | Yes |
| Robust | Yes | Yes | Yes | Yes | Yes | Yes |
| N | 8084 | 8084 | 8084 | 8084 | 8084 | 8084 |
| R2 | 0.215 | 0.291 | 0.240 | 0.310 | 0.231 | 0.299 |

Note: ***, **, and * indicate significance at the 1%, 5% and 10% levels, respectively. The values in parentheses represent t values based on robust standard errors.

financing are insufficient, corporate TFP will be negatively impacted. We expect that the estimated value of the regression coefficient $\beta$ in Model (13) should be significantly negative. That is, the tightening of financing constraints should have a significant negative impact on the company's TFP.

Table 17 reports the estimated results of Model (11), Model (12) and Model (13). The results show that the estimated regression coefficients of the explanatory variable InFin in Model (11) and the explanatory variable Equity in Model (12) are positive at a 1% significance level, and the estimated regression coefficient of the explanatory variable KZ in Model (13) is negative at a 1% significance level, regardless of whether or not the control variables are added. The results are completely consistent with expectations, indicating that the tightening of financing constraints (including the reduction in levels of internal and external equity financing) does have a significant negative impact on the TFP of an underlying company.

The empirical results of Tables 14–17 verify the financing constraint path by which the TFP of listed companies is affected by margin trading in the A-share market.

## Conclusion

Based on the perspective of the stock market information environment and financing constraints of listed companies, we study the impact of the margin trading reform of China's A-share market on the TFP of the underlying companies. We use the gradual expansion of the underlying margin trading stocks in the A-share market as a quasinatural experiment. Through the time-varying PSM-DID method, we find that the margin trading reform in China significantly negatively impacts the TFP of underlying companies. The negative effect is stronger for firms with higher financial leverage, lower cash asset holding levels, lower financial institution holding levels and less securities analyst attention. Further research shows that China's margin trading reform inhibits underlying companies' TFP by worsening the stock market information environment and tightening the financing constraints of listed companies. On the one hand, the resulting increase in asymmetric information makes shareholders tend to use more net profit for cash dividends rather than using internal financing. On the other hand, the increase in asymmetric information also increases a company's external equity financing costs, thereby inhibiting the scale of external equity financing. Our results show how margin trading affects the financing decisions of listed companies by changing the stock market information environment, and ultimately transmitting it to TFP.

This paper combines China's stock market trading system reform with the high-quality development of listed companies to provide a new perspective from which to evaluate China's stock market margin trading reform policy. The results of this paper show that the margin trading reform in China's A-share market does not effectively promote the high-quality development of listed companies. The best practices in optimizing the trading system remain to be further explored.

## Supporting information

**S1 Data.**
(DTA)

## Author Contributions

**Data curation:** Yunqi Ye.

**Formal analysis:** Yunqi Ye.

**Software:** Jingyao Tang.

**Writing – original draft:** Jingyao Tang.

**Writing – review & editing:** Yu Wu.

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
