## [Decision Letter · Decision Letter 0]

23 Mar 2023

PONE-D-23-03601Does Chinese-style Margin Trading Promote the High-quality Development of Listed Companies？PLOS ONE

Dear Dr. Tang,

Thank you for submitting your manuscript to PLOS ONE. After careful consideration, we feel that it has merit but does not fully meet PLOS ONE’s publication criteria as it currently stands. Therefore, we invite you to submit a revised version of the manuscript that addresses the points raised during the review process.

We look forward to receiving your revised manuscript.

Kind regards,

Safdar Husain Tahir, PhD, Postdoc

Academic Editor

PLOS ONE

Journal Requirements:

Additional Editor Comments:

Dear Jingyao Tang

I have been informed that reviewer-2 has recommended major revisions to your manuscript. In order to move forward with the review process, it is important that you address the concerns raised by the reviewer and make the necessary changes to your paper accordingly.

Sincerely,

Reviewers' comments:

Reviewer's Responses to Questions

**Comments to the Author**

1. Is the manuscript technically sound, and do the data support the conclusions?

Reviewer #1: Yes

Reviewer #2: Yes

2. Has the statistical analysis been performed appropriately and rigorously? 

Reviewer #1: Yes

Reviewer #2: Yes

3. Have the authors made all data underlying the findings in their manuscript fully available?

Reviewer #1: Yes

Reviewer #2: Yes

4. Is the manuscript presented in an intelligible fashion and written in standard English?

Reviewer #1: Yes

Reviewer #2: No

5. Review Comments to the Author

Reviewer #1: The Absract meets all requirements.

Introduction has been crafted good.

Literature review integrates theory.

Anaysis is detailed.

Discussion is fine.

I highly appreciate your effort. All I can recommend is proof reading and addition some latest references.

Good luck.

Reviewer #2: This paper explores whether margin trading promotes high quality growth of listed companies, using a quasi-natural experiment of margin trading reform in the Chinese stock market, and provides some insights into the optimization of the institutional environment of the capital market. The statistical analysis is appropriate, and the empirical results support the hypothesis, but the background and the significance of the research question has not been explained clearly. The mechanism should be studied to reveal the effect further.

6. PLOS authors have the option to publish the peer review history of their article (what does this mean?). If published, this will include your full peer review and any attached files.

Reviewer #1: No

Reviewer #2: **Yes: **Xinxin Xu

---

## [Author Response · Author response to Decision Letter 0]

29 Mar 2023

Dear Academic Editor and Reviewers:

Thank you for your letter and the reviewers’ comments on our manuscript entitled "Does Chinese-style Margin Trading Promote the High-quality Development of Listed Companies？" (ID: PONE-D-23-03601). Those comments are very helpful for revising and improving our paper, as well as the important guiding significance to other research. We have studied the comments carefully and made corrections which we hope meet with approval. The main corrections are in the manuscript and the responds to the journal requirements and the reviewers’ comments are as follows.

Response to Journal Requirements

Q1: Please ensure that your manuscript meets PLOS ONE's style requirements, including those for file naming.

Response: We modify our manuscript according to PLOS ONE's style guidelines and articles already published by PLOS ONE. 

Response to Reviews

Reviewer #1: The Absract meets all requirements. Introduction has been crafted good. Literature review integrates theory. Anaysis is detailed. Discussion is fine. I highly appreciate your effort. All I can recommend is proof reading and addition some latest references. Good luck.

Response: Thanks for your helpful recommendations and your high compliment. We are trying our best to make it better. We add some new literatures into paper, as follows:

1. Bennett B, Stulz R, Wang R. Does the stock market make firms more productive?. Journal of Financial Economics. 2020; 136(2): 281-306. doi: 10.2139/ssrn.3080957

2. Chen S, Lee D. Small and vulnerable: SME productivity in the great productivity slowdown. Journal of Financial Economics. 2023; 147: 49-74. doi: 10.1016/j.jfineco.2022.09.007

3. Dai PY, Yang SG, Yuan L. How does stock liquidity affect total factor productivity? On the resource allocation function of the capital market and its development of marketization and legalization. Statistical Research (Chinese). 2022; 39(9): 62-73. https://kns.cnki.net/kcms/detail/detail.aspx?dbname=CJFD2022&filename=TJYJ202209004&dbcode=CJFD

4. Hall BH. The financing of research and development. Oxford Review of Economic Policy. 2002; 18(1): 35-51. doi: 10.1093/oxrep/18.1.35

Reviewer #2: This paper explores whether margin trading promotes high quality growth of listed companies, using a quasi-natural experiment of margin trading reform in the Chinese stock market, and provides some insights into the optimization of the institutional environment of the capital market. The statistical analysis is appropriate, and the empirical results support the hypothesis, but the background and the significance of the research question has not been explained clearly. The mechanism should be studied to reveal the effect further.

Response: Thank you for your wise counsel. Your suggestions have greatly influenced our research and writing.

We made numerous changes to the introduction in order to demonstrate the research background and significance of this paper as clearly as possible (see the first and second paragraphs of the introduction). The support of capital markets is inextricably linked to China's economy's high-quality development. And China's capital market is at a critical stage of reform. Margin trading has been a contentious reform measure in China's stock market in recent years. As a result, the impact of margin trading reform on listed companies' TFP and its mechanism must be clarified. This is not only an intuitive evaluation of the performance of existing reform measures, but also an important reference for future reform measure optimization and improvement. This paper, in particular, closely links the margin trading reform with the TFP of enterprises in theory, based on the stock market information environment and the financing behavior of listed companies. And provides adequate empirical evidence.

In addition, we reorganized the mechanism and presented it succinctly in the introduction. Because of the flaws in reform measures and the unique investment environment of China's stock market, Chinese-style margin trading encourages speculation while discouraging arbitrage. This makes the stock market information environment deteriorate and asymmetric information increase. According to asset pricing theory, the company's market value is reduced, the expected rate of return and the cost of equity capital are increased, and listed companies' external equity financing is hampered. At the same time, according to the principal-agent theory and the dividend signal theory, shareholders have a sufficient incentive to increase cash dividends and reduce retained earnings and internal financing. The concurrent deterioration of internal and external financing will have a negative impact on the TFP of publicly traded companies.

In addition, we have modified and optimized the text expression in other parts of this article based on your suggestions.

We sincerely hope that you will accept our explanation.

---

## [Decision Letter · Decision Letter 1]

13 Apr 2023

Does Chinese-style margin trading promote the high-quality development of listed companies？

PONE-D-23-03601R1

Dear Dr. Tang,

We’re pleased to inform you that your manuscript has been judged scientifically suitable for publication and will be formally accepted for publication once it meets all outstanding technical requirements.

Kind regards,

Safdar Husain Tahir, PhD, Postdoc

Academic Editor

PLOS ONE

Additional Editor Comments (optional):

Reviewers' comments:

Reviewer's Responses to Questions

**Comments to the Author**

1. If the authors have adequately addressed your comments raised in a previous round of review and you feel that this manuscript is now acceptable for publication, you may indicate that here to bypass the “Comments to the Author” section, enter your conflict of interest statement in the “Confidential to Editor” section, and submit your "Accept" recommendation.

Reviewer #2: All comments have been addressed

2. Is the manuscript technically sound, and do the data support the conclusions?

Reviewer #2: Yes

3. Has the statistical analysis been performed appropriately and rigorously? 

Reviewer #2: Yes

4. Have the authors made all data underlying the findings in their manuscript fully available?

Reviewer #2: Yes

5. Is the manuscript presented in an intelligible fashion and written in standard English?

Reviewer #2: Yes

6. Review Comments to the Author

Reviewer #2: (No Response)

7. PLOS authors have the option to publish the peer review history of their article (what does this mean?). If published, this will include your full peer review and any attached files.

Reviewer #2: No

---

## [Editor Report · Acceptance letter]

17 Apr 2023

PONE-D-23-03601R1 

Does Chinese-style margin trading promote the high-quality development of listed companies？ 

Dear Dr. Tang:

I'm pleased to inform you that your manuscript has been deemed suitable for publication in PLOS ONE. Congratulations! Your manuscript is now with our production department. 

Kind regards, 

on behalf of

Dr Safdar Husain Tahir 

Academic Editor

PLOS ONE